# Phylogenetic and divergence analysis of Pentatomidae, with a comparison of the mitochondrial genomes of two related species (Hemiptera, Pentatomidae)

Wang Jia[1], Jing Chen[1], Siyuan Ge[1], Zhenhua Zhang[1], Yuliang Xiao[1], Long Qi[1], Qing Zhao [1]*, Hufang Zhang[2]*

**1** College of Plant Protection, Shanxi Agricultural University, Taigu, Shanxi, China, **2** Department of Biology, Xinzhou Teachers University, Xinzhou, Shanxi, China

\* zhaoqing86623@163.com (QZ); zh_hufang@sohu.com (HZ)

**Data Availability Statement:** All files are available from the NCBI database (accession number: OR500703 and OR500704.)

## Abstract

Pentatomidae, the most diverse family of Pentatomoidea, is found worldwide. Currently, the phylogenetic relationships among Pentatomidae tribes remain unstable, and subfamily divergence has not been estimated. Here, we sequenced and analyzed the complete mitochondrial genomes of two species of *Lelia*, and studied the phylogenetic relationships among Pentatominae tribes. We also selected three available fossil as the calibration points in the family, and preliminarily discussed the divergence time of Pentatomidae. Trees of Pentatomidae were reconstructed using the Bayesian inference method. Divergence times of Pentatominae were estimated based on the nucleotide sequences of protein-coding genes with a relaxed clock log-normal model in BEASTv.1.8.2. The results showed that the gene arrangements, nucleotide composition, and codon preferences were highly conserved in *Lelia*. Further, a phylogenetic analysis recovered Eysarcorini, Strachiini, Phyllocephalini, and Menidini as monophyletic with strong support, however, the monophyly of Antestiini, Nezarini, Carpocorini, Pentatomini and Cappaeini were rejected. Moreover, Pentatominae diverged from Pentatomidae soon after the origin of the Cretaceous Period, at approximately 110.38 Ma. This study enriches the mitochondrial genome database of Pentatomidae and provides a reference for further phylogenetic studies, and provides a more accurate estimate of divergence time.

## Introduction

Pentatomidae, proposed by Leach (1815), is the largest family in the superfamily Pentatomoidea, and it contains 940 genera and 4,949 species in 10 subfamilies [1]. These insects are widely distributed and ubiquitous worldwide. Members of Pentatomidae also vary greatly in size; some are very small, only 2–3 millimeters, such as *Spermatodes variolosus* (Walker, 1867) and *Sepontiella aenea* (Distant, 1883), whereas others are very large, ranging from 20 to 30 millimeters such as *Catacanthus incarnatus* (Drury, 1773). Except for the subfamily Asopinae, which

**Funding:** This research was funded by the National Science Foundation of China (No.31872272); Natural Science Research General Project of Shanxi Province (Nos.2021103021224331); the Research Project Supported by Shanxi Scholarship Council of China (2024-072).

**Competing interests:** The authors have declared that no competing interests exist.

includes predatory species, nearly all species of the Pentatomidae are phytophagous, and many species are found in host plants, whereas some have beenfound to feed on dung or carrion. Owing to a lack of unique methods, the identification of subfamilies and tribes of Pentatomidae, and the construction of a stable taxonomic group have become major problems. In recent years, with the development of sequencing technology, molecular data have been widely used in phylogenetic research [2, 3]. Many researchers have studied the phylogenetic relationships between pentatomids at different classification levels. It is mainly through the sequencing of mitochondrial genome to supplement new molecular data information, build phylogenetic tree, and judge the phylogenetic location of sequenced species and the phylogenetic relationships within some taxa. For example, Yuan et al. (2015) [4] elucidated the phylogenetic relationships among 26 species of Pentatomomorpha based on mitochondrial genomes, demonstrating the monophyly of Pentatomoidea. Lian et al. (2022) [5] newly sequenced mitochondrial genome sequences of three species of the Phyllocephalini and analyzed their phylogenetic position within Pentatomidae. Ding et al. (2023) [6] recently sequenced the mitochondrial genome sequences of three *Menida* species, and clarified the phylogenetic position of *Menida* in Pentatominae. The phylogenetic research of Pentatomidae has always been a hot topic. Grazia et al. (2008) [7] supported the monophyly of Pentatomidae by a combination of morphological and molecular data. However, Roca-Cusaches et al. (2022) [8] used two mitochondrial genes, namely *cox1* and *16S rRNA*, and two molecular genes, *28S rRNA* and *18S rRNA*, to reconstruct a phylogenetic tree of Pentatomidae based on Bayesian Inference (BI) and Maximum Likelihood (ML) methods, and the results denied the monophyly of Pentatomidae. So far, further research is needed on the monophyly and internal relationships of Pentatomidae.

Fossil records provide evidence of the existence of ancient organisms, the most direct and important evidence of biological evolution, but there are also many ambiguous aspects associated with these [9]. Although Pentatomomorpha fossils have been studied for more than 100 years, the number of fossils for this large group remains limited. The earliest known Pentatomomorpha fossils were found in strata for the end of the Late Triassic in Mid west of China and the United Kingdom. Regarding Pentatomomorpha fossils, 14 families, 158 genera and 200 species have been reported [10]. The superfamily Pentatomoidea comprises 18 families worldwide, including two fossil families [1]. However, owing to the incomplete preservation of most fossils and the inability to observe certain key features, it is difficult to fully understand pentatomid origins and evolution [1]. Therefore, it is necessary to use molecular data for such estimations.

The genus *Lelia* was first established by Walker in 1867. Later, Reuter (1890) [11] solved the taxonomic problems associated with some genera, and *Lelia* was used as a valid generic name. Currently, it is a small genus containing only three species worldwide. Moreover, it is widely distributed, and can harm several crops. Members of this genus are typically broad and oval-shaped, and interspecific morphological differences can be identified based on the basal angle of the pronotum and scutellum and the number of spots on the dorsal surface of the body. Fan and Liu (2010) [12] performed a morphological study of this genus with a new species reported. *Lelia decempunctata* (*cox1*) was used to explore the phylogenetic relationships of Pentatomidae [8]. However, to date, a complete mitochondrial genomes of this genus has not been used to explore its phylogenetic relationships and to estimate its of divergence times.

The mitochondrial genomes of insects are double-stranded circular DNA molecules (15–20kb) consisting of 37 genes, specifically 13 protein-coding genes (PCGs), two ribosomal RNA genes (rRNAs), 22 transfer RNA genes (tRNAs), and a control region [13, 14]. In recent years, sequencing technology has developed rapidly, and increasing numbers of insect mitochondrial genomes have been sequenced. Although the functions and replication of the mitochondrial

genome are controlled by the nucleus, due to its stable genetic composition, rapid evolution and relatively complete molecular information, it is still widely used in molecular evolution, phylogeny, population genetic structure and biogeographical research [2, 15].

In this study, we sequenced the whole mitochondrial genomes of two species of *Lelia*, analyzed the mitochondrial genome characteristics in detail, and determined the secondary structures of *12S rRNA* and *16S rRNA*. By comparing and analyzing the mitochondrial genome size, nucleotide composition, codon usage, and RNA structure, we further explored the phylogenetic position of among subfamilies within Pentatomidae. In addition, we used three available Pentatomidae fossil as a fossil calibration point and combined with previous research to estimate the divergence time for each tribe and subfamily. The results of this study will provide a reference for the phylogenetic analysis, identification, origin, and evolution of Pentatomidae.

## Materials and methods

### Sample collection and DNA extraction

Adult specimens of *Lelia concavaemargo* Fan & Liu, 2010 were collected from Baiyan Village, Mashan Town, Meitan County, Zunyi City, Guizhou Province (28˚2′45″N, 107˚35′2″E) on May 10, 2020. Adult specimens of *L. decempunctata* (Motschulsky, 1860) were collected from Yaoluoping Nature Reserve, Yuexi County, Anqing City, Anhui Province (31˚0′56″N, 116˚7′60″E) on July 29, 2019. All samples were immediately placed in anhydrous ethanol and stored in a refrigerator at −25˚C until DNA extraction. Total DNA was extracted from thoracic tissue using a Genomic DNA Extraction Kit (Personalbio, Nanjing, China). The two complete mitogenome were submitted to GenBank (accession numbers: OR500703 and OR500704).

### DNA sequencing, assembly, sequence annotation and analyses

A fluorescent dye (Quant it PicoGreen dsDNA Assay Kit) was used to determine the total amount of DNA. The total amount of DNA was 2.39 mg, and the concentration based on fluorescence was 47.80 ng/ml. The genomic library was constructed using the standard Illumina TruSeq Nano DNA LT library preparation process (Illumina TruSeq DNA Sample Preparation Guide). Whole mitochondrial genome sequencing was performed using an Illumina Novaseq 6000 platform with 400bp insert sizes and a read length of PE150. Fastp was used to evaluate the quality of the sequencing data [16]. Mitochondrial genomes were assembled and annotated using Geneious v. 11.0 software [17]. The reference sequence of *Pentatoma semiannulata* (NC_053653), used for annotation, was obtained from the Basic Local Alignment Search Tool (BLAST) in the NCBI database. The tRNA genes were identified using MITOS (http://mitos.bioinf.uni-leipzig.de/index.py/) with an invertebrate mitochondrial code [18]. The boundaries of the PCGs were determined using the Open Reading Frame Finder on the NCBI website (http://www.ncbi.nlm.nih.gov/gorf/gorf.html). The boundaries of the rRNA genes were identified based on the positions of adjacent genes and previously sequenced rRNA genes [19]. The exact location of the control region was determined based on the boundary of the neighboring genes.

The nucleotide composition and codon usage (RSCU) were analyzed using MEGA v. 11.0 [20]. DnaSP6 software [21] was used to enumerate the non-synonymous substitutions (Ka) and synonymous substitutions (Ks) of each PCG and to calculate the Ka/Ks values. Nucleotide skew was calculated as follows: AT skew = (A −T) / (A + T) and GC skew = (G − C) / (G + C) [18, 22, 23]. The Tandem Repeats Finder web server was used to predict the tandem repeat sequences in the control region [13]. To assess the neutral evolution of species of Pentatominae, we calculated the gene lengths and numbers of non-synonymous and synonymous mutations. Linear regression analyses were performed by comparing 13 PCGs, including the

relationship between the number of non-synonymous mutations and the length of base alignments and the relationship between the number of synonymous mutations and the length of base alignments.

## Phylogenetic analyses

Phylogenetic analyses were conducted using two newly sequenced species and 71 available Pentatomidae taxa from NCBI, with two Tessaratomidae species, *Eusthenes cupreus* (Westwood, 1837) and *Mattiphus splendidus* Distant, 1921, as outgroups (Table 1). The nucleotide sequences of the PCGs and two rRNAs were extracted using Geneious v. 11.0. We further imported the extracted genes into PhyloSuite v.1.2.3 [24], selected MAFFT for the alignment, and used MACSE to optimize the alignment results of the PCGs. We used Gblocks to prune the PCG sequences and TrimAL to prune the rRNA sequences. To determine whether the sequences contained phylogenetic information, we tested the nucleotide substitution saturation and plotted transition and transversion rates against the TN93 distances for the PCGs (all codon positions of the 13 PCGs) and PCGRNA (13 PCGs and two rRNAs) datasets, using DAMBE to further validate the feasibility of constructing a phylogenetic tree [25, 26]. The heterogeneity of sequence divergence in the two datasets was analyzed using AliGROOVE with a default sliding window size [27]. ModelFinder v.2.2.0 was used to provide the best-fit model (S1 Table) [28]. MrBayes v.3.2.7 was used to construct the BI tree [29]. Four independent Markov chains (three heated and one cold) were run for 20,000,000 generations, and samples were collected every 1000 generations. The initial 25% of trees were discarded as burn-in after an average standard deviation of less than 0.01. Phylogenetic trees were constructed using the PCGs and PCGRNA datasets, and the generated phylogenetic trees were visualized using the online editing tool Chipolt [30] (https://www.chiplot.online/).

## Divergence time estimate

Divergence times in Pentatomidae were estimated using the nucleotide sequences of PCGs with a relaxed clock log-normal model in BEAST version 1.8.2 [59]. The PCGs dataset was partitioned using ModelFinder v.2.2.0 [28], and the optimal nucleotide replacement model for each partition was estimated. Appropriate parameters in BEAUti, GTR model, and Yule prior were set for each subset to generate a runnable XML file in BEAST. To estimate the divergence time calibration, *Asopus puncticollis* Piton, 1940 (61.6–59.2 Ma), *Eurydema* Laporte de Castelnau, 1833 (102.24–72.14 Ma) and Pentatomidae (125.0–113.0 Ma), three reported fossils of Pentatomidae, were used to assign the age calibration [60–63]. Tracer v.1.7.2 [64] was used to confirm the convergence of the chain by running the final Markov chain twice every $2\times10^{8}$ generations and sampling every 10,000 generations, with the first 25% of the generations discarded as burn-in. The most effective sample sizes were >200. We used TreeAnnotator v.1.8.4 to obtain the largest branch tree credibility subsample tree. The 95% highest probability density (95%HPD) was displayed using the online editing tool Chipolt [30].

## Results

### Mitochondrial genomic structure

The total lengths of the Pentatomidae mitogenomes were 14,782–19,587 bp, and those of *L. concavaemargo* and *L. decempunctata* were 16,074 and 15,464 bp respectively (Fig 1). The mitogenomes of the two species were determined to be closed circular double-stranded DNA molecules containing 37 genes (13 PCGs, 22 tRNAs, and two rRNAs) and a control region. The arrangement of the mitochondrial genome was consistent with 23 genes located on the J-

**Table 1. List of species used to construct the phylogenetic tree.**

| Family | Subfamily | Tribe | Species | GenBank number | Reference |
|---|---|---|---|---|---|
| Pentatomidae | Asopinae | | *Arma custos* | NC_051562 | [31] |
| | | | *Arma koreana* | OP902493 | Gao et al. [Unpublished] |
| | | | *Cazira horvathi* | NC_042817 | [32] |
| | | | *Cazira verrucosa* | OP920754 | Zhao et al. [Unpublished] |
| | | | *Dinorhynchus dybowskyi* | NC_037724 | [33] |
| | | | *Eocanthecona furcellata* | MZ440302 | [34] |
| | | | *Eocanthecona thomsoni* | NC_042816 | [32] |
| | | | *Picromerus lewisi* | NC_058610 | [35] |
| | | | *Picromerus griseus* | NC_036418 | [36] |
| | | | *Picromerus viridipunctatus* | OP920756 | Zhao et al. [Unpublished] |
| | | | *Stiretrus anchorago* | BK059217 | [37] |
| | | | *Zicrona caerulea* | NC_058303 | [38] |
| | Pentatominae | Aelini | *Aelia sibirica* | NC_067883 | Zhao et al. [Unpublished] |
| | | Aelini | *Aelia fieberi* | NC_067750 | Chen et al. [Unpublished] |
| | | Aeschrocorini | *Aeschrocoris ceylonicus* | OP526368 | [39] |
| | | Aeschrocorini | *Aeschrocoris tuberculatus* | OP526367 | [39] |
| | | Antestiini | *Anaxilaus musgravei* | NC_061538 | Zhao et al. [Unpublished] |
| | | Antestiini | *Plautia stali* | NC_072252 | Lin et al. [Unpublished] |
| | | Antestiini | *Plautia lushanica* | NC_058973 | [40] |
| | | Antestiini | *Plautia fimbriata* | NC_042813 | [32] |
| | | Antestiini | *Plautia crossota* | NC_057080 | [41] |
| | | Cappaeini | *Homalogonia obtusa* | NC_070404 | Gao et al. [Unpublished] |
| | | Cappaeini | *Halyomorpha halys* | NC_013272 | [42] |
| | | Carpocorini | *Euschistus heros* | BK059218 | [37] |
| | | Carpocorini | *Dolycoris baccarum* | NC_020373 | [43] |
| | | Catacanthini | *Catacanthus incarnatus* | NC_042804 | [32] |
| | | Caystrini | *Hippotiscus dorsalis* | NC_058969 | [40] |
| | | Caystrini | *Caystrus obscurus* | NC_042805 | [32] |
| | | Eysarcorini | *Eysarcoris rosaceus* | MT165687 | [44] |
| | | Eysarcorini | *Eysarcoris montivagus* | MW846867 | [44] |
| | | Eysarcorini | *Eysarcoris guttigerus* | NC_047222 | [45] |
| | | Eysarcorini | *Eysarcoris gibbosus* | MW846868 | [44] |
| | | Eysarcorini | *Eysarcoris annamita* | MW852483 | [44] |
| | | Eysarcorini | *Eysarcoris aeneus* | MK841489 | [46] |
| | | Eysarcorini | *Carbula sinica* | NC_037741 | [47] |
| | | Halyini | *Dalpada cinctipes* | NC_058967 | [40] |
| | | Halyini | *Erthesina fullo* | NC_042202 | [48] |
| | | Hoplistoderini | *Hoplistodera incisa* | NC_042799 | [32] |
| | | Menidini | *Menida violacea* | NC_042818 | [32] |
| | | Menidini | *Menida musiva* | OP066239 | [6] |
| | | Menidini | *Menida metallica* | OP066240 | [6] |
| | | Menidini | *Menida lata* | OP066241 | [6] |
| | | Myrocheini | *Tholosanus proximus* | NC_063300 | Zhao et al. [Unpublished] |
| | | Nezarini | *Palomena viridissima* | NC_050166 | [49] |
| | | Nezarini | *Palomena angulosa* | NC_068746 | Gao et al. [Unpublished] |
| | | Nezarini | *Nezara viridula* | NC_011755 | [50] |
| | | Nezarini | *Glaucias dorsalis* | NC_058968 | [40] |

(*Continued*)

**Table 1.** (Continued)

| Family | Subfamily | Tribe | Species | GenBank number | Reference |
|---|---|---|---|---|---|
| | | Pentatomini | *Placosternum urus* | NC_042812 | [32] |
| | | Pentatomini | *Pentatoma semiannulata* | NC_053653 | [51] |
| | | Pentatomini | *Pentatoma rufipes* | MT861131 | [52] |
| | | Pentatomini | *Pentatoma metallifera* | NC_058972 | [40] |
| | | Pentatomini | *Neojurtina typica* | NC_058971 | [40] |
| | | Pentatomini | **Lelia concavaemargo** | OR500703 | This study |
| | | Pentatomini | **Lelia decempunctata** | OR500704 | This study |
| | | Piezodorini | *Piezodorus guildinii* | BK059215 | [37] |
| | | Sciocorini | *Sciocoris lateralis* | NC_072066 | Liu et al. [Unpublished] |
| | | Sephelini | *Brachymna tenuis* | NC_042802 | [32] |
| | | Strachiini | *Eurydema ventralis* | MG584837 | Zhao et al. [Unpublished] |
| | | Strachiini | *Eurydema qinlingensis* | NC_044765 | [53] |
| | | Strachiini | *Eurydema oleracea* | NC_044764 | [53] |
| | | Strachiini | *Eurydema maracandica* | NC_037042 | [54] |
| | | Strachiini | *Eurydema liturifera* | NC_044763 | [53] |
| | | Strachiini | *Eurydema gebleri* | NC_027489 | [4] |
| | | Strachiini | *Eurydema dominulus* | NC_044762 | [53] |
| | Phyllocephalinae | Phyllocephalini | *Chalcopis glandulosa* | ON991494 | [5] |
| | | Phyllocephalini | *Dalsira scabrata* | NC_037374 | [47] |
| | | Phyllocephalini | *Gonopsimorpha nigrosignata* | ON991492 | [5] |
| | | Phyllocephalini | *Gonopsis affinis* | NC_036745 | [55] |
| | | Phyllocephalini | *Gonopsis coccinea* | ON991493 | [5] |
| | Podopinae | Deroploini | *Deroploa parva* | NC_063299 | Zhao et al. [Unpublished] |
| | | Tarisini | *Dybowskyia reticulata* | NC_070271 | Zhao et al. [Unpublished] |
| | | Graphosomatini | *Graphosoma rubrolineatum* | NC_033875 | [56] |
| | | Podopini | *Scotinophara lurida* | NC_042815 | [32] |
| Tessaratomidae | | | *Eusthenes cupreus* | NC_022449 | [57] |
| | | | *Mattiphus splendidus* | NC_053743 | [58] |

strand, and 14 genes on the N-strand. Moreover, the longest intergenic spacers (22 and 20 bp) of the two species were detected between *nad1* and *trnS2*, and the longest overlapping region was located between *trnW* and *trnC* and had a length of 8 bp (AAGCTTTA). In addition, there were two conserved overlaps, with a 7 bp overlap between *atp8*/*atp6* and *nad4*/*nad4L* (ATGATAA) (S2 Table). This has also been observed in other species of the Pentatomidae family. The nucleotide composition of the total mitogenomes showed a strong bias toward A and T bases; further, the AT skew was positive and the GC skew was negative, with A+T contents of 78.14% (*L. concavaemargo*) and 77.73% (*L. decempunctata*) (S3 Table).

## Protein-coding genes

Similar to those of other Pentatomidae members, the nucleotide compositions of the 13 PCGs of these two species had high AT contents, specifically 77.34% (*L. concavaemargo*) and 77.38% (*L. decempunctata*), respectively (S3 Table). In these two species, nine genes were found to be encoded on the major strand (J-strand), whereas the other four were encoded on the minor strand (N-strand). Most PCGs used ATN (ATT/ATA/ATG/ATC) as the initiation codon, but some PCGs (*cox1*, *atp8*, *nad6*, *nad1*) used TTG as the initiation codon. Most PCGs ended with

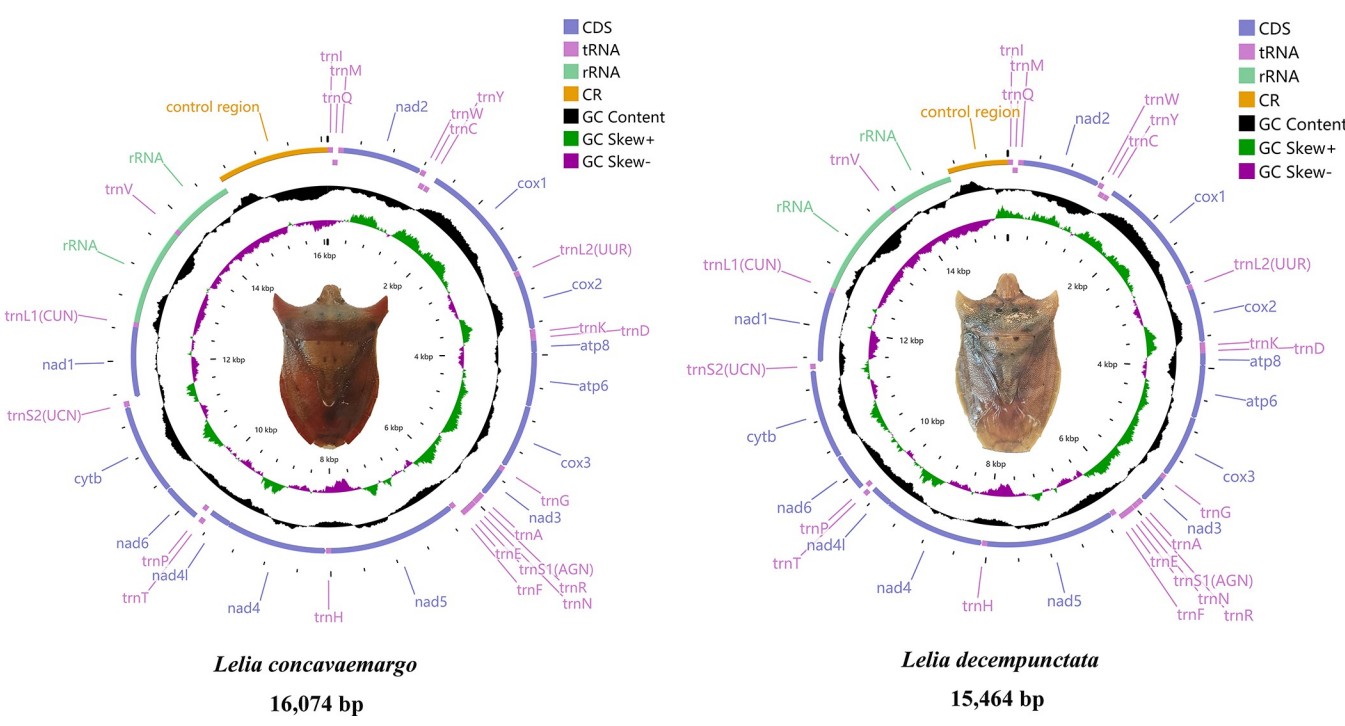

**Fig 1. Gene arrangements of the two complete mitochondrial genomes.**

the complete termination codon TTA, except for *cox1* and *cox2*, which ended with the incomplete stop codon T (S2 Table).

We also calculated the RSCU of PCGs for both species and a similar RSCU pattern was observed (Fig 2). Most of the codons with high frequency ended in A/T, and the most

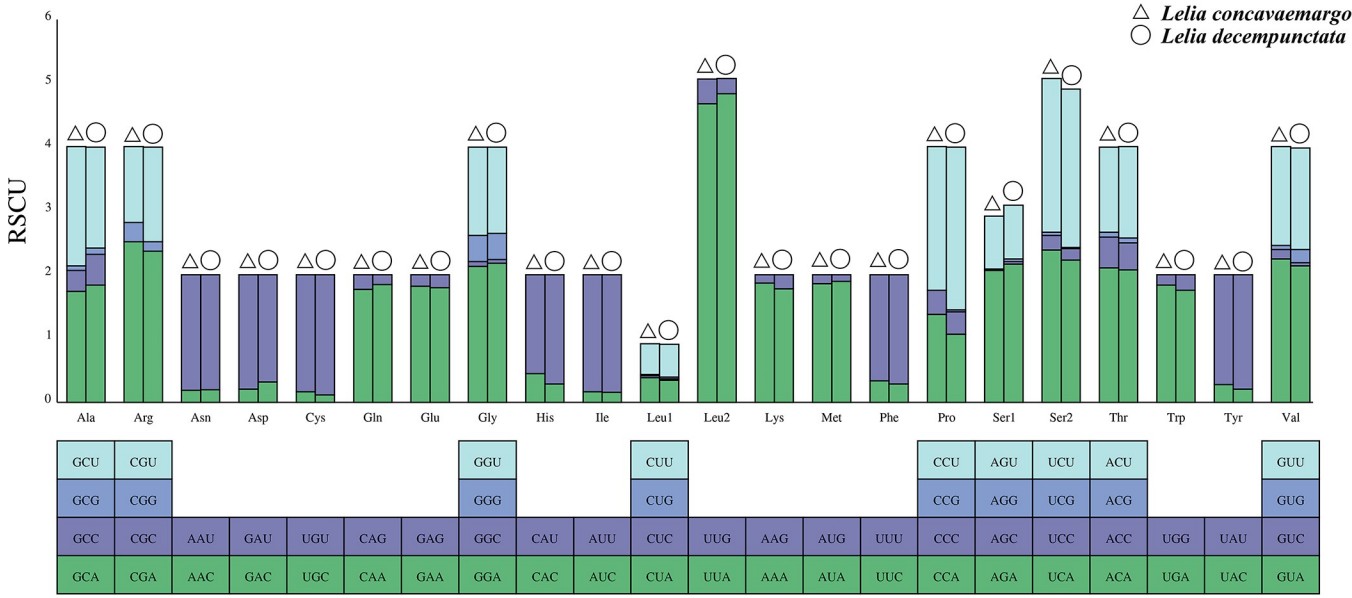

**Fig 2. Relative synonymous codon usage (RSCU) within *L. concavaemargo* and *L. decempunctata*.** Codon families are shown on the x-axis and the frequency of RSCU on the y-axis.

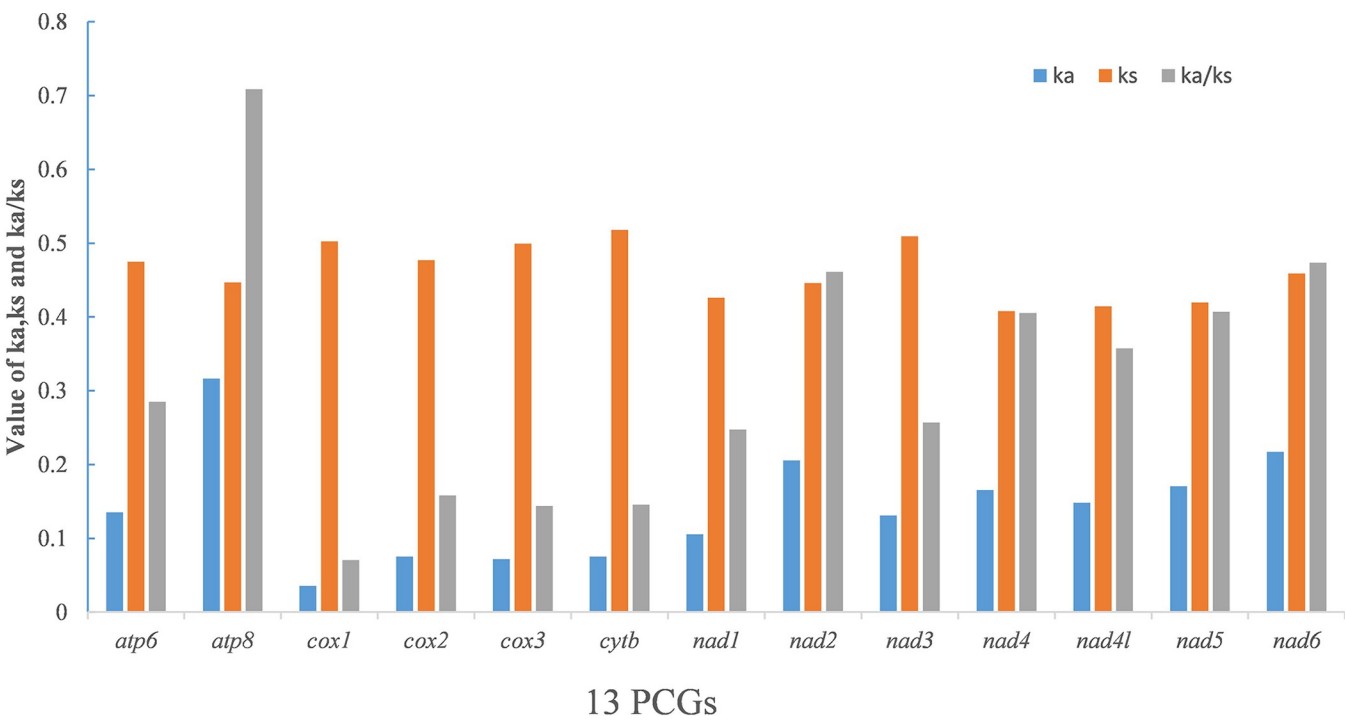

**Fig 3. Evolutionary rates of 13 PCGs in Pentatomidae.** Rate of non-synonymous substitutions (Ka), rate of synonymous substitutions (Ks), and ratio of rate of non-synonymous substitutions to rate of synonymous substitutions (Ka/Ks) are calculated for each PCG.

frequently used codon was UUA (Leu2). These results indicated that the codons of the PCGs of *Lelia* tended to end with A/T.

In addition, we determined the Ka, Ks, and Ka/Ks ratios for the 13 PCGs of Pentatomidae to explore evolutionary patterns. The Ka/Ks ratio for all 13 PCGs was < 0.71, indicating that these genes were affected by purifying selection. Among the PCGs, *atp8* evolved at the fastest rate (Ka/Ks = 0.71), whereas *cox1* evolved at the slowest rate (Ka/Ks = 0.07) (Fig 3). Owing to its slow evolution rate, we determined that *cox1* can be used for barcode analysis and classification. Linear regression analysis showed that non-synonymous and synonymous changes were significantly correlated with the gene length ($R^2$ = 1.000, 0.996) (Fig 4).

## Transfer and ribosomal RNAs

The total tRNA lengths of *L. concavaemargo* and *L. decempunctata* were 1,479 bp and 1,481 bp, respectively (S3 Table). The 22 tRNAs showed high A + T contents of 78.30% (*L. concavaemargo*) and 78.12% (*L. decempunctata*), and the lengths of the tRNA genes ranged from 63 bp to 75 bp. Fourteen genes were located on the J-strand, and eight other genes were located on the N-strand (S2 Table). Only *trnS1* and *trnV* lacked the dihydrouridine (DHU) arm, and the remaining 20 tRNA genes formed a typical cloverleaf structure in both species. However, in most Pentatomidae species, only *trnS1* lacked the DHU arm. *trnR* showed the weakest conservation compared to the other genes in the two species of *Lelia*. Moreover, 17 wobble G-U pairs were found in 22 tRNAs from *Lelia* (Fig 5).

The total lengths of the two rRNAs were 2,106 bp (*L. concavaemargo*) and 2,107 bp (*L. decempunctata*). Moreover, they were encoded on the N-strand and showed high AT contents of 79.63% (*L. concavaemargo*) and 79.83% (*L. decempunctata*) (S3 Table). The complete

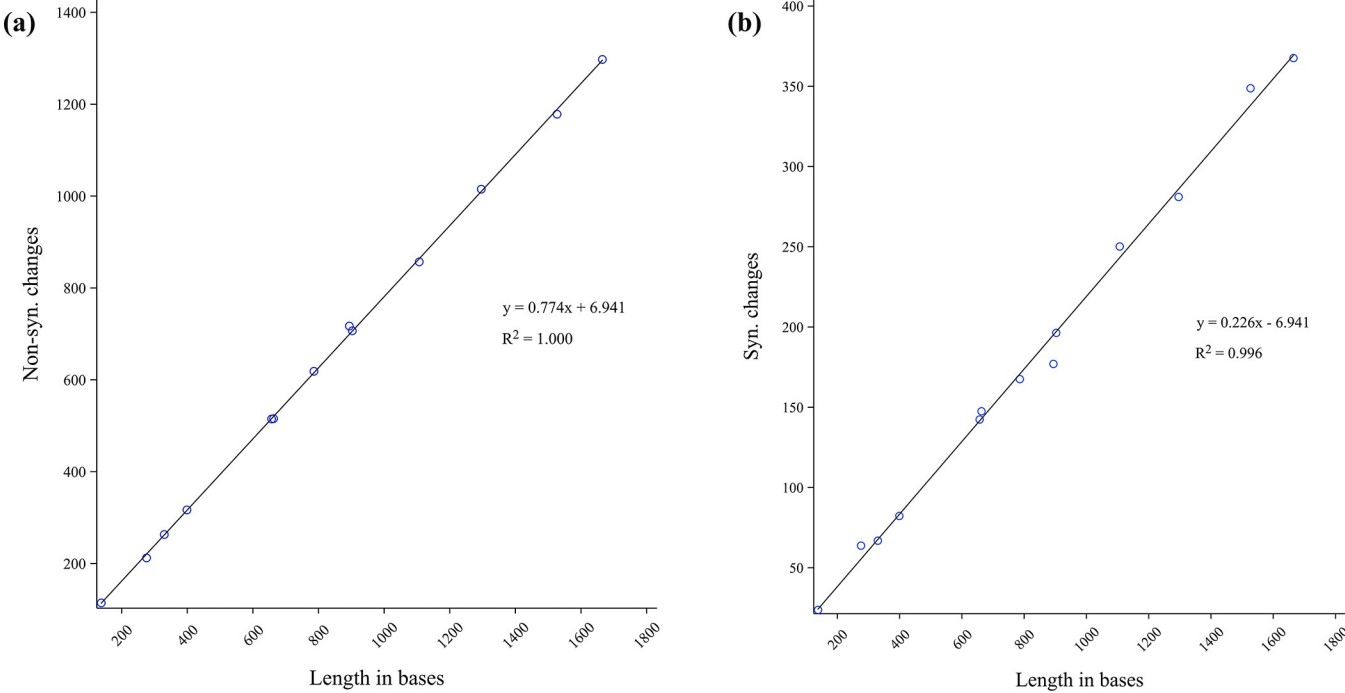

**Fig 4.** (a) Correlation between nonsynonymous mutations and length in bases of the genes. (b) Correlation between synonymous mutations and length in bases of the genes.

secondary structures are shown in Figs 6 and 7. In *Lelia*, *16S rRNA* contained 92.12% conserved sites and *12S rRNA* contained 93.97% conserved sites.

## Control ergion

The control region was determined to be the largest non-coding region. In the mitochondrial genome of Pentatomidae, the longest control region was 4,651 bp, and there was a significant difference in length between these two species, with lengths of 1,378 bp (*L. concavaemargo*) and 762 bp (*L. decempunctata*). Moreover, the control regions were located between *12S rRNA* and *trnI*. High AT contents of 81.28% (*L. concavaemargo*) and 73.75% (*L. decempunctata*) (S3 Table). In *L. concavaemargo*, five types of tandem repeat units were observed, with lengths of 3–68 bp, however, in *L. decempunctata*, only one type of tandem repeat unit was found, with a length of 56 bp (S4 Table).

## Phylogenetic relationships

We next analyzed the substitution saturation and heterogeneity of the PCGs and PCGRNA dataset, before constructing a phylogenetic tree. The results showed that the Xia saturation index was below the critical values for a symmetric and asymmetric topologies (Iss < Iss.c, $p < 0.05$) (Fig 8), indicating that the nucleotide sequences of the two datasets were not saturated. The heterogeneity between both sequences is shown in blue, and the lightest part of the blue occurred between the outgroup and the remaining sequences in the dataset (Fig 9), indicating that these datasets are suitable for further phylogenetic studies.

We also constructed phylogenetic trees for Pentatomidae based on two datasets (PCGs and PCGRNA) using BI (Fig 10 and S1 Fig). The results showed that the topological structures of the two trees were reliable and that most clades had high posterior probabilities. A topology of

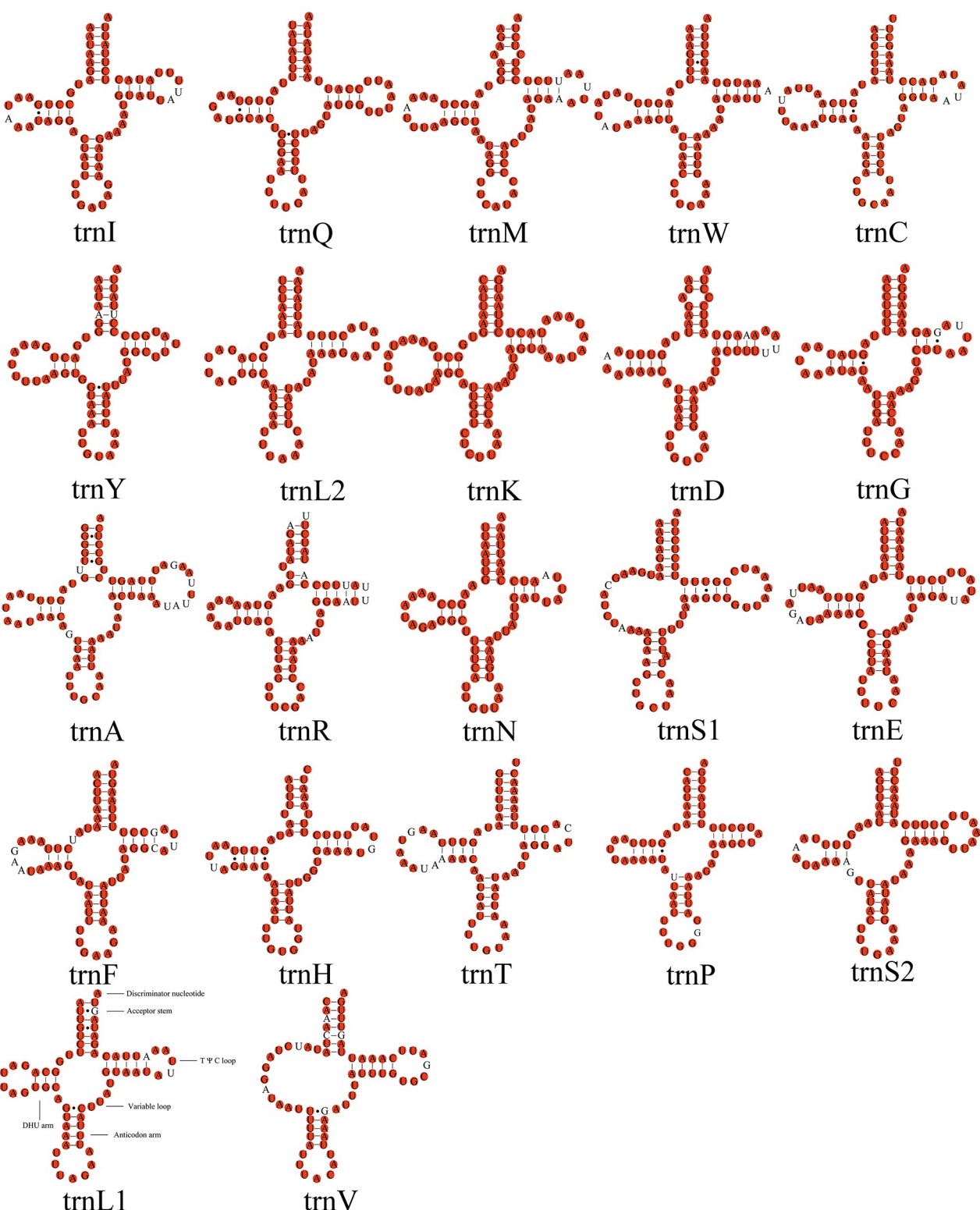

**Fig 5. Predicted secondary structure of tRNA genes in *L. concavaemargo*.** The conserved sites within *Lelia* are marked in red.

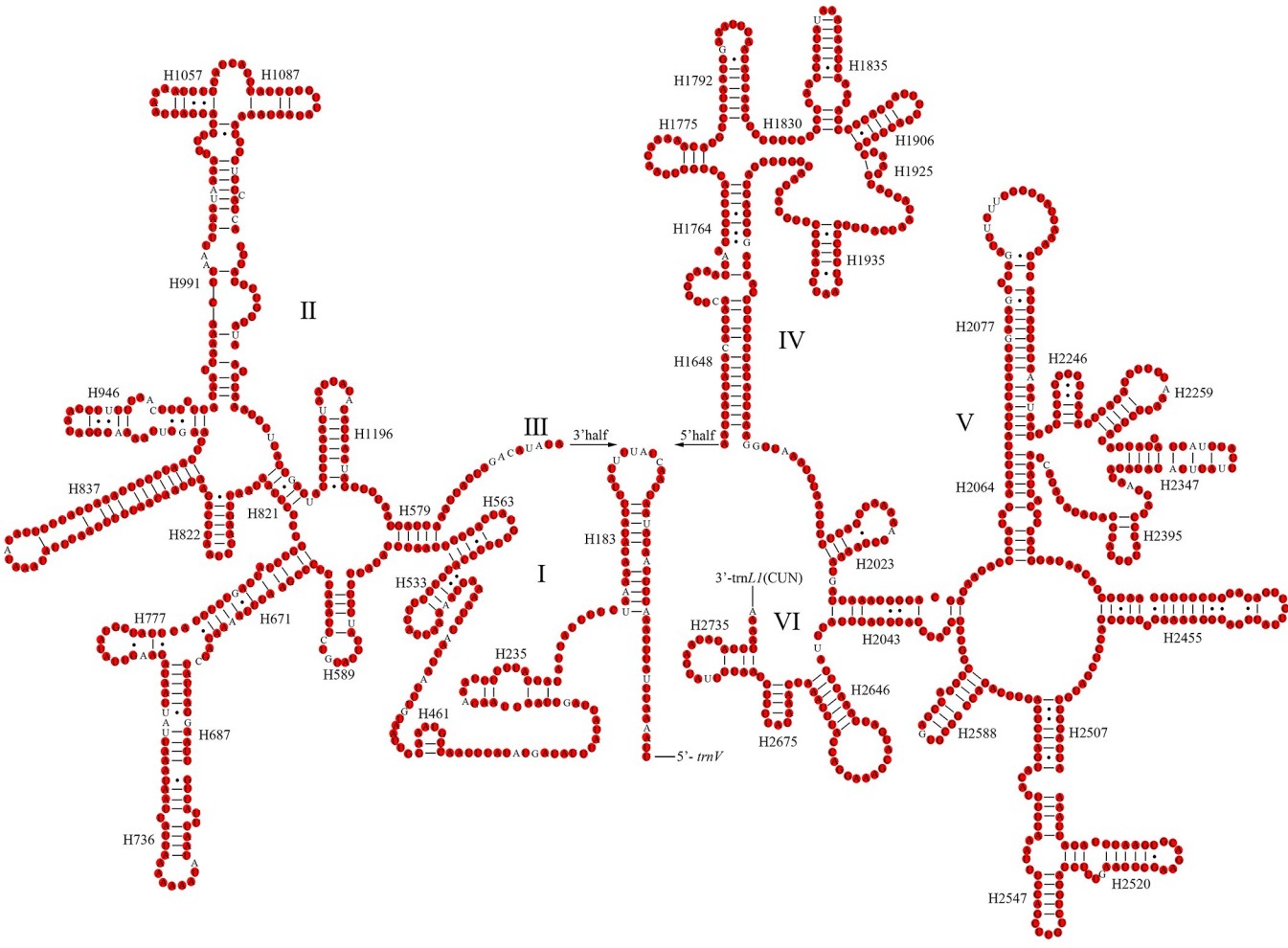

**Fig 6. Predicted secondary structure of the *16S rRNA* in *L. concavaemargo*.** The conserved sites within *Lelia* are marked in red.

PCG dataset is as follows: (Aeschrocorini + (*Neojurtina* + (*Euschistus* + ((*Sciocoris* + (*Graphosoma* + *Dybowskyia*)) + ((Caystrini + (*Homalogonia* + Halyini)) + (*Halyomorpha* + (*Placosternum* + Phyllocephalini)))) + (Nezarini + (*Anaxilaus* + (*Glaucias* + Antestiini))) + ((*Dolycoris* + Aelini) + Eysarcorini) + ((Piezodorini + *Brachymna*) + ((*Deroploa* + *Tholosanus*) + Pentatomini)) + (*Hoplistodera* + Strachiini) + ((*Catacanthus* + *Scotinophara*) + (Menidini + Asopinae))))))))). The topologies showed that Aeschrocorini was the earliest diverging lineage within Pentatomidae. Eysarcorini, Strachiini, and Menidini were recovered as monophyletic with strong support; however, the monophyly of Antestiini, Nezarini, Carpocorini, Pentatomini, Cappaeini, and Podopinae was rejected. Owing to the limited mitochondrial genomic data available in the NCBI database, the monophyly of the remaining tribes could not be accurately verified.

## Divergence time estimate

Based on the topology of Pentatomidae recovered from the BEAST analysis, the age estimates, average, and 95% HPD for each subfamily and tribe are summarized in Fig 11. We used three type fossil information within the Pentatomidae to analyze and update the divergence time of family. *Asopus puncticollis* and *Eurydema* Laporte, as crown groups, and Pentatomidae as a

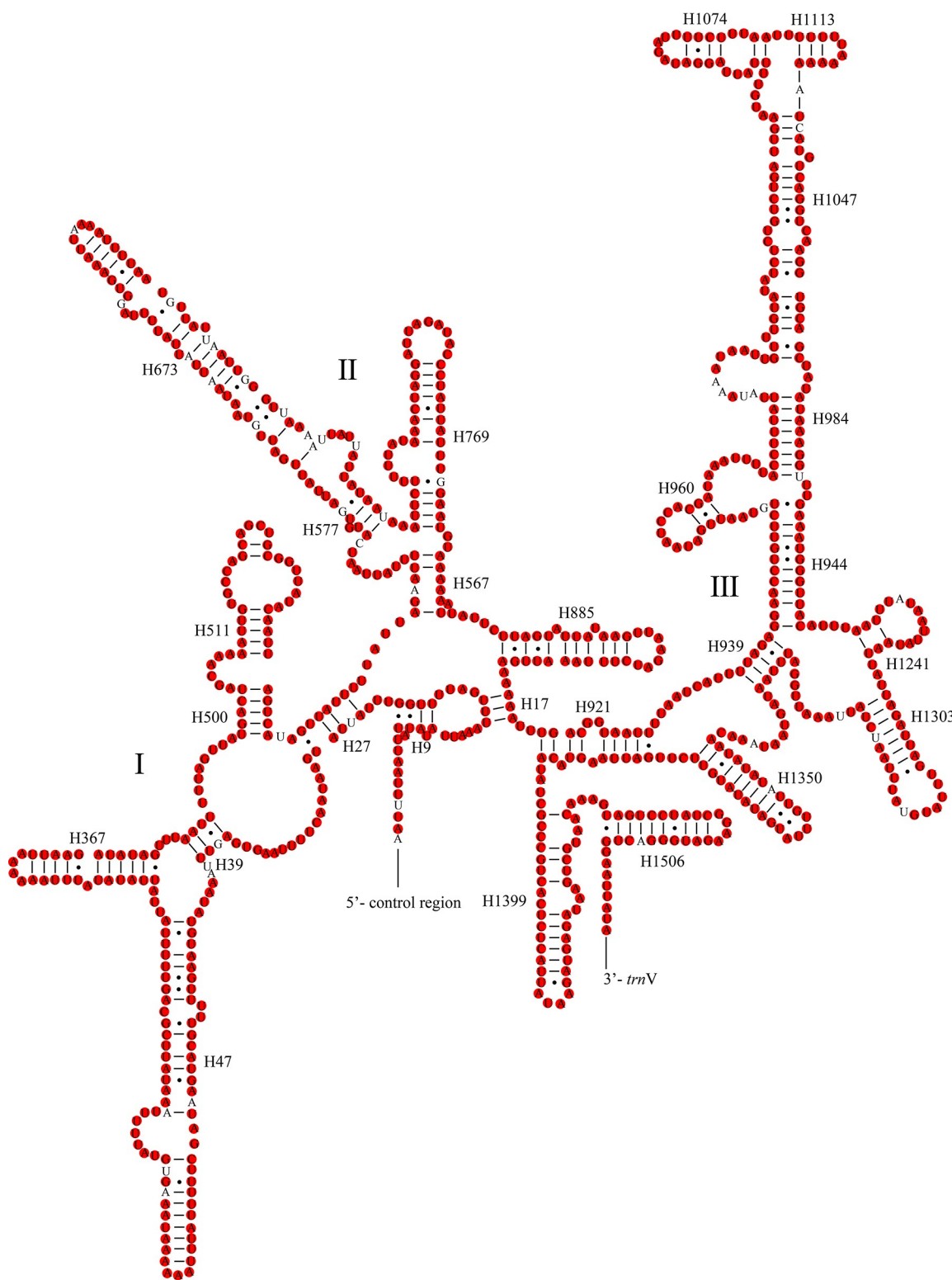

**Fig 7. Predicted secondary structure of the *12S rRNA* in *L. concavaemargo*.** The conserved sites within *Lelia* are marked in red.

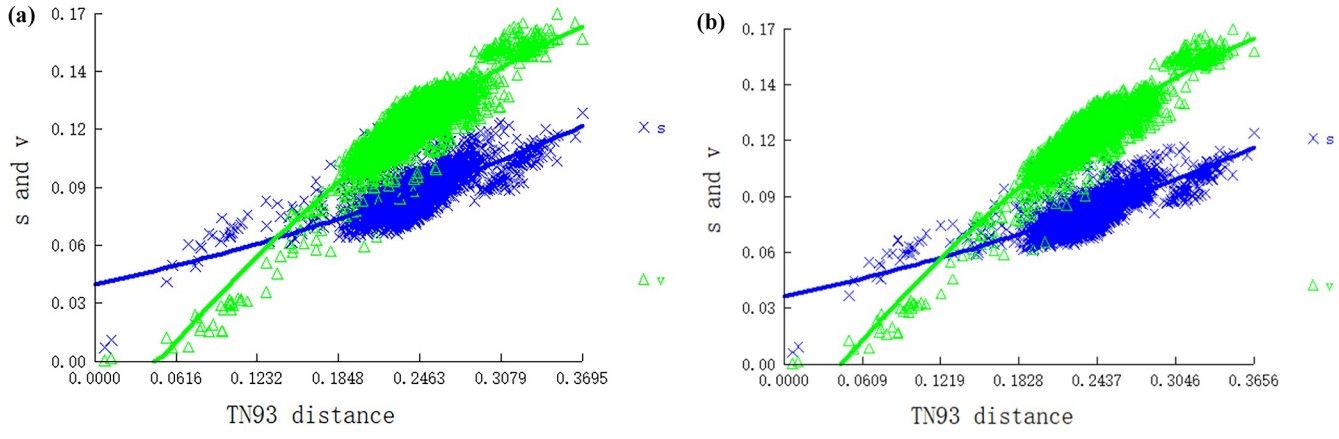

**Fig 8.** The substitution saturation analysis of two datasets (a) PCGs (b) PCGRNA.

stem groups showed that the divergence of Pentatomidae occurred from the Cretaceous to the Quaternary period.

Further, Pentatominae diverged from Pentatomidae soon after the origin of the Cretaceous period, at approximately 110.38 Ma (95% HPD: 138.38–83.77 Ma). After a period of evolution, Aeschrocorini and *N. typica* were the earliest to diverge, with a divergence time of 89.15 Ma (95% HPD: 109.75–68.95 Ma), which occurred in the Upper period of the Cretaceous. The divergence time of *Lelia* and *Pentatoma* was 46.81 Ma (95% HPD: 58.94–35.53 Ma), which occurred in the Eocene and Paleocene periods of the Paleogene. The divergence time of Phyllocephalinae was 67.60 Ma (95% HPD: 82.56–51.76 Ma), which occurred in the Upper Campanian period of the Cretaceous to the Eocene Ypresian period of the Paleogene. *Scotinophara lurida* was the first to differentiate in Podopinae, and this occurred 62.66 Ma (95% HPD: 77.29–47.61 Ma) in the Upper Campanian period of the Cretaceous to Eocene Lutetian period of the Paleogene. The divergence time of Asopinae and Menidini was 68.49 Ma (95% HPD: 77.29–47.61 Ma), which occurred in the Eocene Ypresian period of the Paleogene to the Upper Campanian period of the Cretaceous.

## Discussion and conclusions

In this study, we compared two mitochondrial genomes, and the results showed that the gene arrangement was consistent with other published mitochondrial genomes of Pentatomidae [33, 42, 57].

The length of the control region was found to be closely related to the number of tandem repeat units. Other pentatomid species with different length of control regions and tandem repeats have been reported in previous studies [4, 44]. This finding strongly suggests that the length of the control region determines the length of the entire mitochondrial genes.

Similar to that observed in other pentatomid species, the mitochondrial genomes of the two species of *Lelia* exhibited a preference for an asymmetric nucleotide composition is thought to be caused by mutational pressure and natural selection [23]. Generally, in PCGs, *cox1* is widely used in taxonomic studies of insects because it is a potential marker for species identification [65, 66]. The fact that the number of synonymous and non-synonymous mutations was highly correlated with the length of their respective genes was evidence of neutral evolution, which is consistent with that previously predicted for the mitochondrial genome [67].

In *Lelia*, the lack of DHU arm in *trnS1* (AGN) is a typical feature of insect mitogenomes [68–70]. Apart from typical Watson–Crick pairings (G-C and A-U), some atypical G-U

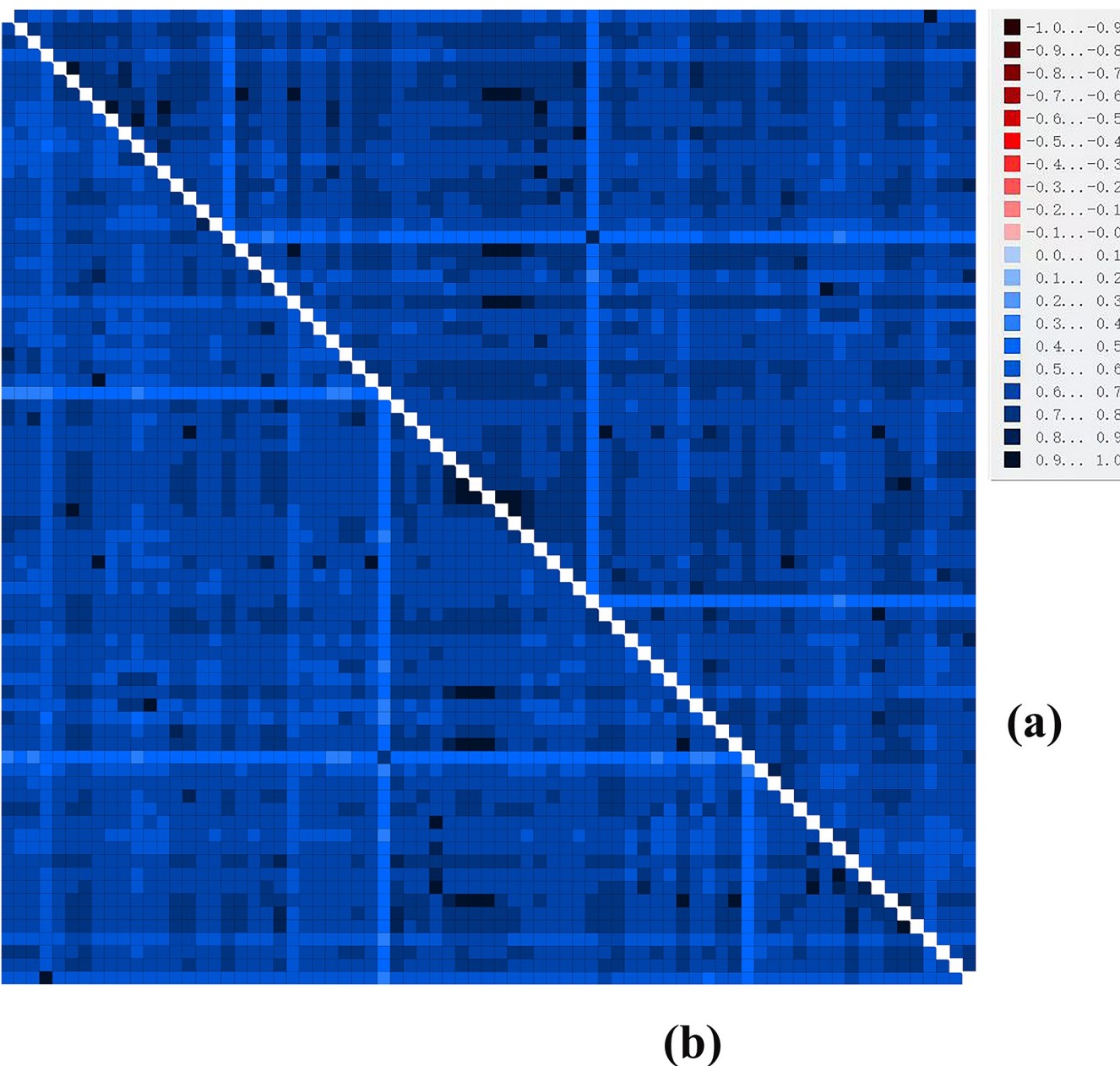

**Fig 9.** Analysis of heterogeneity of sequence divergence for (a) PCGs and (b) PCGRNA dataset. The mean similarity score between sequences is represented by colored squares, based on AliGROOVE scores ranging from –1, which indicates a great difference in rates from the remainder of the data set (= heterogeneity, red color) to +1, which indicates rates that matched all other comparisons (blue color, as in this case).

pairings can be transformed into fully functional proteins via post-transcriptional mechanisms [71, 72].

The phylogenetic results were consistent with those of traditional morphological research [1]. *Lelia* and *Pentatoma* have a close genetic relationship but very different morphologies. Moreover, the species of tribe Aeschrocorini was the first species to differentiate from Pentatomidae, and previous studies have reported similar result [39]. The phylogenetic trees constructed based on the two datasets formed the same topology on branch, as follow: (Nezarini + (*Anaxilaus* + (*Glaucias* + Antestiini))) and ((*Dolycoris* + Aelini) + Eysarcorini) (Fig 10 and S1

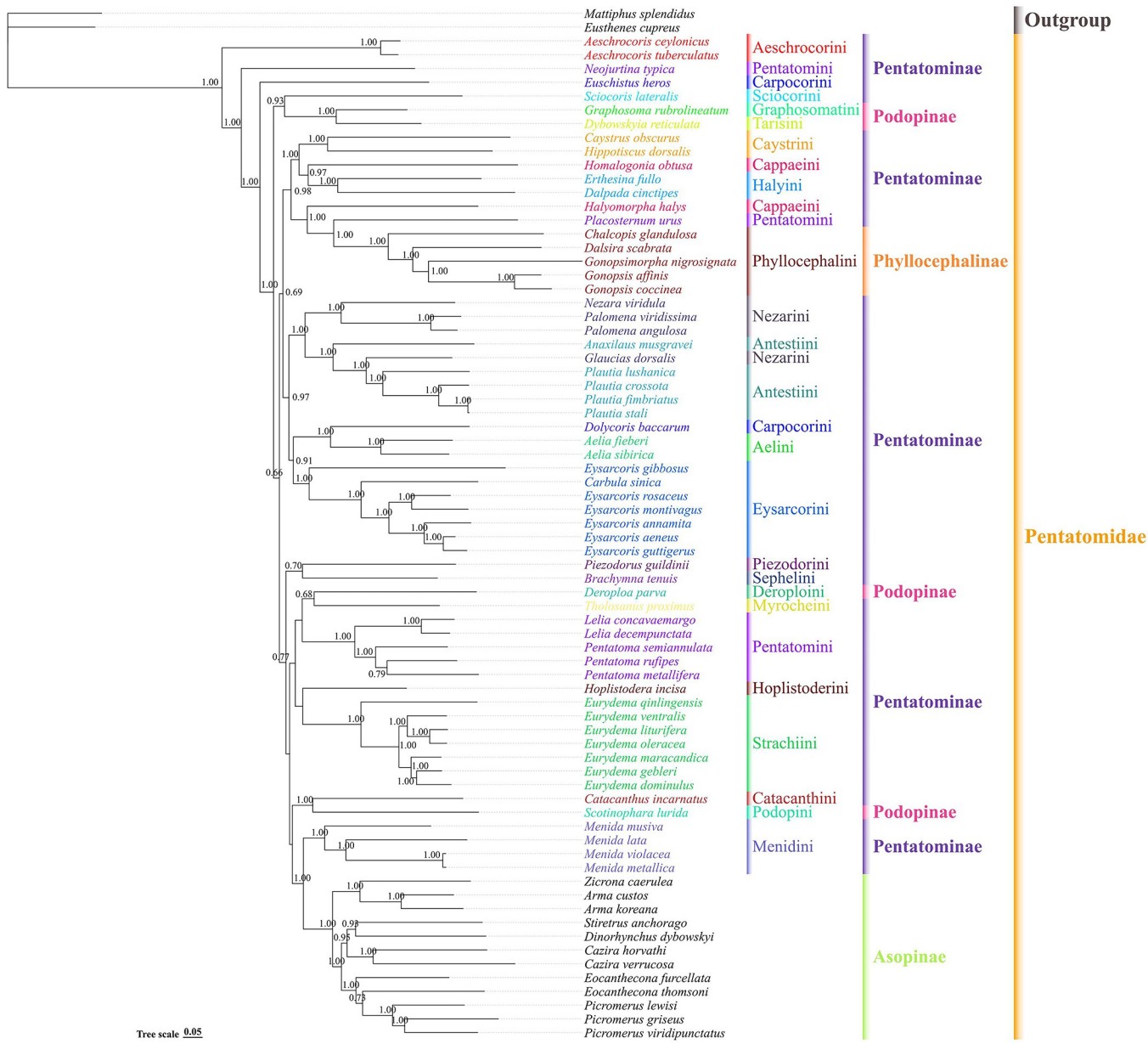

**Fig 10. The phylogenetic relationships of tribes within Pentatomidae reconstructed from DNA sequences of 13 protein coding using BI methods.**
Numbers on nodes are the posterior probabilities (PP), lower than 0.6 is not displayed.

Fig). Antestiini and Nezarini are closely related but do not form a monophyletic group. However, because of the uncertainty of its location, *Plautia* Stål, 1864 has been temporarily placed in the Antestiini tribe; moreover the placement of the genus *Plautia* has been problematic and could bridge the gap between this tribe and Nezarini [1]. In Eysarcorini, *Eysarcoris gibbosus* (Jakovlev, 1904) was the first to diverge from other species; our results support the suggestion of Roca-Cusachs and Jung (2019) and Li et al. (2021) to transfer *E. gibbosus* to *Stagonomus* Gorski, 1852 [44, 73]. Morphologically, Eysarcorini and Carpocorini are extremely similar [1], yet the molecular data that we examined indicate that these two tribes are closely related. In previous studies, the sister group relationship between Menidini and Asopinae has been

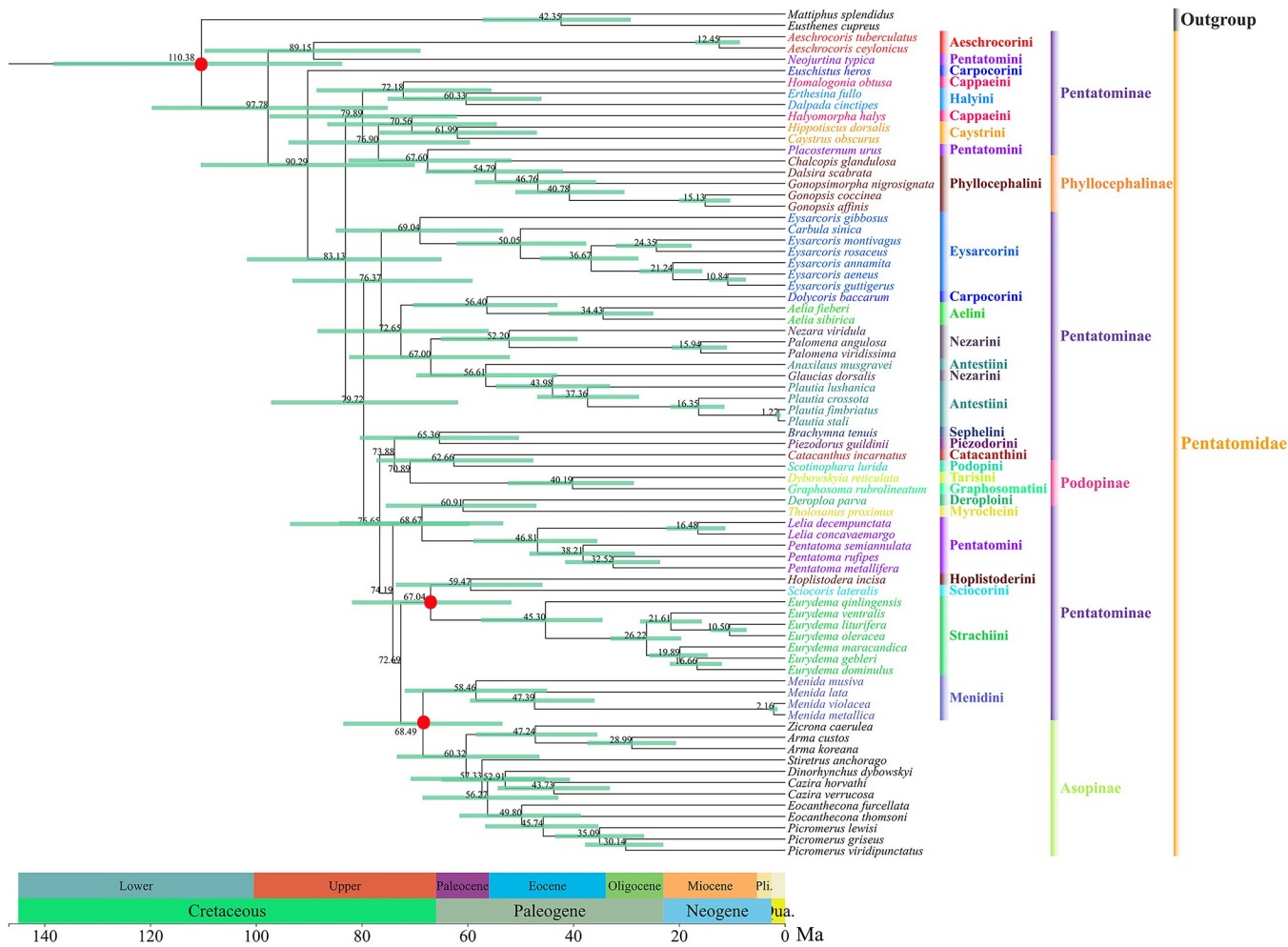

**Fig 11. The chronogram of divergence times within Pentatomidae by BEAST v.1.8.4 analysis.** Horizontal bars represent 95% credibility intervals of time estimates. Numbers on the nodes indicate the mean divergence times. The calibration point is the red dot in the figure.

supported through nuclear and mitochondrial gene analyses [8]. In the phylogenetic tree that we constructed, owing to limited data and differences in the results between the two datasets, it was difficult to analyze the phylogenetic relationships and classification status of the remaining tribes.

This study was the first to estimate the divergence time of various tribes of the lower taxon Pentatomidae. We selected three fossils that are currently available as the calibration points for Pentatomidae. The earliest divergence of Pentatomidae occurred in the Lower Cretaceous, which is consistent with previous research [62]. The emergence of Asopinae indicates that the transition of their feeding habits from herbivorous to predatory is closely related to their divergence time. Except for Pentatominae, the divergence times of the other three subfamilies are relatively close (Fig 11). Moreover, previous studies have focused on higher taxa for divergence time estimations. In the absence of fossil evidence, we collected fossils within Pentatomidae as the calibration point, and these had a closer genetic relationship than that observed in previous studies on higher-order elements, making the research results more reliable; at the same time, the estimation of the divergence time of each branch node was more accurate, resulting in higher research significance.

In conclusion, our study not only examined the genus *Lelia* at the molecular level and identified its taxonomic position in phylogenetic relationships but also discussed the subfamily and tribe evolution in Pentatomidae. We also provide a theoretical basis for the evolutionary history of Pentatomidae. It is necessary to sequence more mitochondrial genomes and to discover more fossil data for further studies.

## Supporting information

**S1 Table. Partitions and models based on model finder of PCGs and PCGRNA.**
(XLSX)

**S2 Table. Organization of the mitochondrial genomes of *L. concavaemargo* and *L. decempunctata.***
(XLSX)

**S3 Table. Nucleotide composition of the mitogenomes of *L. concavaemargo* and *L. decempunctata.***
(XLSX)

**S4 Table. Tandem repeats of the control region of the mitochondrial genomes of *L. concavaemargo* and *L. decempunctata.***
(XLSX)

**S1 Fig. The phylogenetic relationships of tribes within Pentatomidae reconstructed from DNA sequences of 13 protein coding and 2 rRNA mitochondrial genes using BI methods.** Numbers on nodes are the posterior probabilities (PP), lower than 0.6 is not displayed.
(JPG)

## Acknowledgments

Many thanks to Xiaofei Ding and Dan Lian for collecting the material specimens, and thanks Editage for linguistic assistance during the preparation and revision of this manuscript.

## Author Contributions

**Conceptualization:** Wang Jia, Qing Zhao.

**Data curation:** Wang Jia, Jing Chen.

**Formal analysis:** Wang Jia, Jing Chen, Siyuan Ge.

**Investigation:** Wang Jia, Zhenhua Zhang.

**Methodology:** Wang Jia, Siyuan Ge, Yuliang Xiao.

**Project administration:** Qing Zhao, Hufang Zhang.

**Resources:** Siyuan Ge, Long Qi.

**Supervision:** Wang Jia, Qing Zhao.

**Validation:** Wang Jia, Qing Zhao.

**Writing – original draft:** Wang Jia, Jing Chen, Siyuan Ge.

**Writing – review & editing:** Qing Zhao.

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
