## [Decision Letter · Decision Letter 0]

14 May 2024

PONE-D-24-13214Phylogenetic and divergence analysis of Pentatomidae, with a comparison of the mitochondrial genomes of two related species (Hemiptera, Pentatomidae)PLOS ONE

Dear Dr. Zhao,

Thank you for submitting your manuscript to PLOS ONE. There are few important concerns highlighted by both the reviewers independently and I feel that these concern should be addressed before the manuscript can be accepted formally. Therefore, we invite you to submit a revised version of the manuscript that addresses the points raised during the review process. Please submit your revised manuscript by Jun 28 2024 11:59PM. If you will need more time than this to complete your revisions, please reply to this message or contact the journal office at plosone@plos.org. Please include the following items when submitting your revised manuscript:A rebuttal letter that responds to each point raised by the academic editor and reviewer(s). You should upload this letter as a separate file labeled 'Response to Reviewers'.A marked-up copy of your manuscript that highlights changes made to the original version. You should upload this as a separate file labeled 'Revised Manuscript with Track Changes'.An unmarked version of your revised paper without tracked changes. You should upload this as a separate file labeled 'Manuscript'.

We look forward to receiving your revised manuscript.

Kind regards,

Pankaj Bhardwaj, Ph.D.

Academic Editor

PLOS ONE

Journal Requirements:

"This research was funded by the National Science Foundation of China (No.31872272); Key Forestry Research and Development Plan of Shanxi Province (LYZDYF2023-35)；Natural Science Research General Project of Shanxi Province (Nos.202103021224331)."

"The authors declare no conflict of interest. "

4. We note that you have referenced (Unpublished) on page 8, which has currently not yet been accepted for publication. Please remove this from your References and amend this to state in the body of your manuscript: (ie “Bewick et al. [Unpublished]”) as detailed online in our guide for authors

Reviewers' comments:

Reviewer's Responses to Questions

**Comments to the Author**

1. Is the manuscript technically sound, and do the data support the conclusions?

Reviewer #1: Yes

Reviewer #2: Yes

2. Has the statistical analysis been performed appropriately and rigorously? 

Reviewer #1: Yes

Reviewer #2: N/A

3. Have the authors made all data underlying the findings in their manuscript fully available?

Reviewer #1: Yes

Reviewer #2: Yes

4. Is the manuscript presented in an intelligible fashion and written in standard English?

Reviewer #1: Yes

Reviewer #2: Yes

5. Review Comments to the Author

Reviewer #1: The manuscript presents the mitogenome features, phylogeny and divergence time estimation in the family Pentatomidae based on 2 newly and 71 available Pentatomidae taxa from NCBI. The study is important for the area because this family has not been evaluated by this much broad sampling at the family and/or subfamily level comparing the monophyletic relationships with the mitophylogenomics of the family so far. But there are some questions need to be clarified and improved:

1. The manuscript needs to more discussion on the estimated divergence time between genera/subfamilies.

2. Figures and tables or supporting information should be cited adequately in the section of Discussion for better interpretation.

3. Please consider the all abbreviations across the manuscript.

4. Some parts of the manuscript have been repeated and the same explanation is also included in the result/discussion section. It therefore needs to be shortened and discussed more comprehensively considering the relevant literature.

Reviewer #2: The authors reported the complete mitochondrial genomes of two Lelia species within the family Pentatomidae and reconstructed phylogeny with their closely related taxa.

Major comments:

This manuscript needs extensive English editing. Some paragraphs and sentences need to be completely rewritten.

The Introduction section should add more sentences to describe the phylogenetic conflicts or somewhat reflect your contributions in this work.

I suggest that the description of mitochondrial genome structure and characteristics be appropriately reduced in the Discussion section

Minor comments:

Line 25: "Pentatomidae, the most diverse subfamily of Pentatomoidea"; however, " Pentatomidae" is a family name, not subfamily.

Line 27: Caution in the use of "controversial" in phylogenetic studies.

Line 31: "divergence time of Pentatomidae" instead of "Pentatomidae divergence time"; "Trees" instead of "trees"; "reconstructed" instead of "constructed".

Line 32: "Divergence time of Pentatomidae"

Line 40: Delete "(95% highest probability density: 138.38-83.77 Ma)", this information is not required to be included in the Abstract section.

Lines 42-43: How do you think your results were more accurate?

Line 48: superfamily Pentatomoidea.

Lines 50-53: You described the various body sizes of pentatomid members; however, I do not understand the connection between body size and mitogenomes or phylogeny.

Line 62: "nuclear genes" instead of "molecular genes".

Lines 60-76: Should be completely rewritten. You listed many previous works but just emphasize the monophyly of Pentatomidae in some studies, is it because its monophyly has not yet been determined? What is the connection between these works and your focused taxa in your study?

Lines 67 and 75: Rephase the statement of "between the family Pentatomidae and Phyllocephalini" and "between Pentatominae and Menida". They are not the same level in taxonomy.

Line 167: "PhyloSuite".

Lines 297-300: Phylogenetic trees were reconstructed based on two datasets, and only one tree was selected as your main phylogenetic hypothesis. I noticed that the topologies of these two trees were distinct. Try to explain the reason why you selected the "PCG" topology.

Lines 310-311: Do you think the non-monophyly is due to the limited mitogenomic data?

6. PLOS authors have the option to publish the peer review history of their article (what does this mean?). If published, this will include your full peer review and any attached files.

Reviewer #1: No

Reviewer #2: No

---

## [Author Response · Author response to Decision Letter 0]

17 Jul 2024

Dear Editor, 

Thanks for your letter and for reviewers’ comments concerning our manuscript entitled “Phylogenetic and divergence analysis of Pentatomidae, with a comparison of the mitochondrial genomes of two related species (Hemiptera, Pentatomidae)”. These comments are all valuable and helpful for revising our manuscript. We have read through comments carefully and have made corrections. Revisions in the text are shown using red highlight for additions, and strikethrough font for deletions. The responses to the reviewer's comments are marked in red and presented following.

Thank you for reviewing again and we look forward to hearing from you.

Sincerely,

Qing Zhao

First of all, we made a statement on the following points.

1. We have ensured that the manuscript format conforms to the style of plos one.

3. The authors have declared that no competing interests exist.

4. We have revised and proofread the references on page 8 according to your suggestions.

Response to Reviewer #1

Point 1: The manuscript needs to more discussion on the estimated divergence time between genera/subfamilies.

Response: Thank you for the reviewer's comments. We have added further discussion on the estimated divergence time between genera/subfamilies in the manuscript.

Point 2: Figures and tables or supporting information should be cited adequately in the section of Discussion for better interpretation.

Response: Thanks to the reviewer's suggestion, we added the citation in the corresponding position in the manuscript.

Point 3 ： Please consider the all abbreviations across the manuscript.

Response: Thanks to your suggestions. We double-checked all the abbreviations in the manuscript.

Point 4： Some parts of the manuscript have been repeated and the same explanation is also included in the result/discussion section. It therefore needs to be shortened and discussed more comprehensively considering the relevant literature.

Response: Thank you for the reviewer's comments. We have shortened and revised the discussion section.

Response to Reviewer #2

Major comments:

Point 1: This manuscript needs extensive English editing. Some paragraphs and sentences need to be completely rewritten.

Response: Thanks to your suggestion, we have rewritten some paragraphs and sentences in the manuscript.

Point 2: The Introduction section should add more sentences to describe the phylogenetic conflicts or somewhat reflect your contributions in this work.

Response: Thank you for your advice. In the Introduction, we have added Pentatomidae phylogenetic conflicts of previous researchers' findings, and described the content and contributions of the entire manuscript study in the last paragraph.

Point 3: I suggest that the description of mitochondrial genome structure and characteristics be appropriately reduced in the Discussion section.

Response: Thanks to your suggestions, I have appropriately reduced the description of mitochondrial genome structure and characteristics in the Discussion section.

Minor comments:

Point 1: Line 25: "Pentatomidae, the most diverse subfamily of Pentatomoidea"; however, " Pentatomidae" is a family name, not subfamily.

Response: Thank you for pointing out the problem, I have corrected it.

Point 2: Line 27: Caution in the use of "controversial" in phylogenetic studies.

Response: Thanks to your suggestions, I replaced “controversial” with “unstable” in the manuscript.

Point 3: Line 31: "divergence time of Pentatomidae" instead of "Pentatomidae divergence time"; "Trees" instead of "trees"; "reconstructed" instead of "constructed".

Response: Thanks to your suggestions, I have corrected it in the manuscript.

Point 4: Line 32: "Divergence time of Pentatomidae"

Response: Thank you, I have corrected it in the manuscript.

Point 5: Line 40: Delete "(95% highest probability density: 138.38-83.77 Ma)", this information is not required to be included in the Abstract section.

Response: Thanks to your suggestions, I deleted it in the Abstract section of the manuscript.

Point 6: Lines 42-43: How do you think your results were more accurate?

Response: We selected all currently applicable fossil information of Pentatomidae, including the calibration of stem group nodes and crown group nodes, to further study the divergence time within Pentatomidae more accurately. 

Point 7: Line 48: superfamily Pentatomoidea.

Response: Thanks for your suggestion, I have corrected it in the manuscript.

Point 8: Lines 50-53: You described the various body sizes of pentatomid members; however, I do not understand the connection between body size and mitogenomes or phylogeny.

Response: Thank you for your question. In the manuscript, we describe the various body sizes of pentatomid members, focusing on the rich diversity of the species, followed by a phylogenetic analysis based on the use of mitochondrial genome data.

Point 9: Line 62: "nuclear genes" instead of "molecular genes".

Response: Thanks for your suggestion, I have corrected it in the manuscript.

Point 10: Lines 60-76: Should be completely rewritten. You listed many previous works but just emphasize the monophyly of Pentatomidae in some studies, is it because its monophyly has not yet been determined? What is the connection between these works and your focused taxa in your study?

Response: Thanks for your suggestion, I have corrected Lines 60-76 in the manuscript. This study elucidates the phylogenetic position of Lelia, and further research is needed on the monophyly of Pentatomidae.

Point 11: Lines 67 and 75: Rephase the statement of "between the family Pentatomidae and Phyllocephalini" and "between Pentatominae and Menida". They are not the same level in taxonomy.

Response: Thank you for your suggestion. I have rewritten this section.

Point 12: Line 167: "PhyloSuite".

Response: Thank you for pointing out the problem, I have corrected it.

Point 13: Lines 297-300: Phylogenetic trees were reconstructed based on two datasets, and only one tree was selected as your main phylogenetic hypothesis. I noticed that the topologies of these two trees were distinct. Try to explain the reason why you selected the "PCG" topology.

Response: We consider the length of the whole manuscript and select the “PCG” topology as a representative to further construct the divergence time tree based on the PCG dataset.

Point 14: Lines 310-311: Do you think the non-monophyly is due to the limited mitogenomic data?

Response: I think the limited mitogenomic data will affect the judgment of monophyly. Animal mitochondrial genome has many common characteristics, such as stability of gene composition, relatively conservative gene arrangement, common maternal inheritance, and rarely recombination[1], so it has been widely used in evolutionary and phylogenetic studies[2]. The limited mitochondrial genome data may not be fully representative of the genetic diversity of the study subjects. This may affect the judgment of monophyly in phylogenetic results.

Reference:

[1] Wolstenholme D R. Animal mitochondrial DNA: structure and evolution. International review of cytology, 1992, 141: 173-216.

[2] Wilson A C, Cann R L, Carr S M, et al. Mitochondrial DNA and two perspectives on evolutionary genetics. Biological Journal of the Linnean Society, 1985, 26(4): 375-400.

---

## [Decision Letter · Decision Letter 1]

15 Aug 2024

Phylogenetic and divergence analysis of Pentatomidae, with a comparison of the mitochondrial genomes of two related species (Hemiptera, Pentatomidae)

PONE-D-24-13214R1

Dear Dr. Zhao,

We’re pleased to inform you that your manuscript has been judged scientifically suitable for publication and will be formally accepted for publication once it meets all outstanding technical requirements.

Kind regards,

Pankaj Bhardwaj, Ph.D.

Academic Editor

PLOS ONE

Additional Editor Comments (optional):

Reviewers' comments:

Reviewer's Responses to Questions

**Comments to the Author**

1. If the authors have adequately addressed your comments raised in a previous round of review and you feel that this manuscript is now acceptable for publication, you may indicate that here to bypass the “Comments to the Author” section, enter your conflict of interest statement in the “Confidential to Editor” section, and submit your "Accept" recommendation.

Reviewer #1: All comments have been addressed

Reviewer #3: (No Response)

2. Is the manuscript technically sound, and do the data support the conclusions?

Reviewer #1: Yes

Reviewer #3: Yes

3. Has the statistical analysis been performed appropriately and rigorously? 

Reviewer #1: Yes

Reviewer #3: Yes

4. Have the authors made all data underlying the findings in their manuscript fully available?

Reviewer #1: Yes

Reviewer #3: Yes

5. Is the manuscript presented in an intelligible fashion and written in standard English?

Reviewer #1: Yes

Reviewer #3: Yes

6. Review Comments to the Author

Reviewer #1: I have no idea in this round. The authors addressed all question that I have provide. I recommend this manuscript publish in PLOS One.

Reviewer #3: In this manuscript, authors sequenced and analyzed the complete mitochondrial genomes of two species of Lelia, and studied the phylogenetic relationships among Pentatominae tribes. This study enriches the mitochondrial genome database of Pentatomidae and has important value for further phylogenetic research. I think it is worthy to be published in PLOS ONE.

This manuscript can be accepted after the minor issue have been modified：

In CONCLUSION: “We also provide a theoretical basis for the evolutionary history of Pentatomidae.”

I think this conclusion is too exaggerated, and I believe that this study has not fundamentally analyzed the evolutionary history of Pentatomidae. This sentence should be deleted or modified in its expression.

7. PLOS authors have the option to publish the peer review history of their article (what does this mean?). If published, this will include your full peer review and any attached files.

Reviewer #1: No

Reviewer #3: No

---

## [Editor Report · Acceptance letter]

21 Aug 2024

PONE-D-24-13214R1 

PLOS ONE

Dear Dr. Zhao, 

I'm pleased to inform you that your manuscript has been deemed suitable for publication in PLOS ONE. Congratulations! Your manuscript is now being handed over to our production team.

Kind regards, 

on behalf of

Dr. Pankaj Bhardwaj 

Academic Editor

PLOS ONE